biomathematics

emergence, smallpox, epidemic,
mathematical model, oscillation, data

**Author for correspondence:**
Meredith Greer
e-mail: mgreer@bates.edu

# Emergence of oscillations in a simple epidemic model with demographic data

Meredith Greer[1], Raj Saha[2], Alex Gogliettino[3],
Chialin Yu[3] and Kyle Zollo-Venecek[3]

[1]Department of Mathematics, Bates College, Lewiston, Maine 04240, USA
[2]Department of Geology and Department of Physics and Astronomy, Bates College, Lewiston, Maine 04240, USA
[3]Bates College, Lewiston, Maine 04240, USA

 MG, 0000-0002-9543-4498

A simple susceptible–infectious–removed epidemic model for smallpox, with birth and death rates based on historical data, produces oscillatory dynamics with remarkably accurate periodicity. Stochastic population data cause oscillations to be sustained rather than damped, and data analysis regarding the oscillations provides insights into the same set of population data. Notably, oscillations arise naturally from the model, instead of from a periodic forcing term or other exogenous mechanism that guarantees oscillation: the model has no such mechanism. These emergent natural oscillations display appropriate periodicity for smallpox, even when the model is applied to different locations and populations. The model and datasets, in turn, offer new observations about disease dynamics and solution trajectories. These results call for renewed attention to relatively simple models, in combination with datasets from real outbreaks.

## 1. Introduction

Mathematical models for disease reappearance across time in a specified geographical area typically incorporate delay terms, age structure or periodic forcing. These modeller-selected features permit fine-tuned adjustment of oscillations for the purpose of matching a dataset, and periodic forcing terms, in particular, can permit the teasing apart of different contributors to disease incidence. Still, a chicken-and-egg question must be asked: should observed oscillations in the data cause us to construct a model in which periodic behaviour is guaranteed, or should a model written without the expectation of periodicity be permitted to display oscillation, or not, depending on its parameter values and underlying demographics?

More questions closely follow: what can we learn from sustained oscillations in a very simple model? What is the role of historical data in determining model parameters? What might we discover about disease transmission or population dynamics by parametrizing a model with demographic data? And how do results from a mathematical model help to highlight unusual or especially interesting data points from the historical record?

We address these questions using historical smallpox and demographic data from three geographically separated regions: the Hida district of Japan, British India and Sweden. We begin with an overview of modelling history and results, then employ a series of computational approaches that constrain parameter values for each of the three regions by comparing model, data, and observed outbreak periods. We next circle back to the data, demonstrating the ability of a calibrated model to illuminate unusual historical occurrences. Throughout, we show that oscillation is an emergent property of simple models involving demographics.

## 2. History and overview of modelling approaches

Periodic outbreaks have piqued the interest of researchers for over two hundred years. In 1929, statistician Soper [1] commented on the periodicity of measles outbreaks. He applied a second-order differential equation model, of the kind regularly used for mass-spring systems, to represent oscillations and connect with data. Soper refers [1] to work approximately twenty years earlier by epidemiologist Sir William Hamer. Hamer later writes about Noah Webster's 1799 volume *A Brief History of Epidemic and Pestilential Diseases* [2] as saying that influenza is a 'transitory scourge only afflicting the peoples of the world at times separated by long intervals of freedom' [3].

Current research more typically draws from the classic 1927 paper by Kermack & McKendrick [4] that introduced compartmental models, now often known by initials such as SIR for 'susceptible-infectious-removed'. Sustained oscillations in differential equations-based SIR and related models are frequently described using delay differential equations, periodic forcing terms involving sine or cosine functions, and/or age structure. The marvellous 2000 review paper by Hethcote [5] describes seven articles that model sustained oscillation in epidemics, and all seven use one or more of these approaches. It is also frequently the case that modellers keep total population size constant. The sustained oscillation discussion in Hethcote [5] does not refer to models with non-constant population size.

The 2008 text *Mathematical Epidemiology* [6] features two chapters that thoroughly describe the topic of oscillation in compartmental models. In Chapter 1, Earn provides references to the analysis showing that a fixed-population-size SIR model with constant parameter values displays damped oscillations toward a stable endemic equilibrium. As a follow-up, he uses [7] to show that stochasticity in model demographics—a staple in any real-world dataset—can sustain oscillations indefinitely. In Chapter 11, Bauch describes models having sustained oscillations as involving either exogenous or endogenous mechanisms. An exogenous mechanism, typically periodic forcing of the transmission term, explicitly introduces oscillations into the model. Endogenous mechanisms instead destabilize the model's endemic equilibrium. The most frequent examples of endogenous mechanisms are age structure or delay terms. Other examples include stochastic approaches, or a non-constant contact rate of infectious and susceptible individuals.

Before and since *Mathematical Epidemiology* summarized these findings, both exogenous and endogenous mechanisms have continued to be incorporated into models. Networks [8,9] and cellular automata [10–12] provide additional ways to understand oscillations in SIR models. Each new approach extends the concept of the compartmental model in a way that yields new insight.

By contrast, we step back, invoking Occam's Razor and asking what can be learned from a simpler model. Such a model keeps the transmission parameter constant, along with all other parameters that directly represent aspects of smallpox: the basic reproduction number $\mathcal{R}_0$, length of infectivity, percentage of deaths due to smallpox. Only the parameters for overall births and non-smallpox deaths in the population vary, their values determined by annual computations based on historical demographic data, with annual values interpolated using straight lines to prevent discontinuous inputs to the system of differential equations. Periodic or quasi-periodic oscillations from such a reduced and simple model can then point to the fundamental ingredients of oscillatory dynamics in disease outbreaks.

In addition to this model in which the parameters for births and non-smallpox deaths vary with time—hereafter referred to as the demographically forced model—we present a corresponding autonomous model. A formula for oscillation period in the autonomous model is analytically determined, and parameter values for the autonomous model are obtained through a combination of wavelet analysis and simulation-based experiments. Ultimately, the analytically determined formula

for oscillation period, when computed with the obtained parameter values, is shown to be consistent with demographic data. Overall this study demonstrates that (i) a simple dynamical model with time-varying births and deaths is sufficient to explain sustained periodicity of outbreaks, and (ii) demographic and disease outbreak data can be used to constrain model parameters such as the infection parameter, $\beta$.

## 3. Model description

We work with the autonomous model

$$
\left.\begin{aligned}
\frac{dS}{dt} &= \alpha - \beta SI - \delta S, \\
\frac{dI}{dt} &= \beta SI - (\gamma + \epsilon)I = \beta SI - \mu I \\
\frac{dR}{dt} &= \gamma I - \delta R
\end{aligned}\right\}
\tag{3.1}
$$

and

and its demographically forced counterpart

$$
\left.\begin{aligned}
\frac{dS}{dt} &= \alpha(t) - \beta SI - \delta(t)S, \\
\frac{dI}{dt} &= \beta SI - (\gamma + \epsilon)I = \beta SI - \mu I \\
\frac{dR}{dt} &= \gamma I - \delta(t)R,
\end{aligned}\right\}
\tag{3.2}
$$

and

where $S$, $I$ and $R$ are the susceptible, infectious and recovered populations. The parameter $\beta$ is the transmission coefficient, $\alpha$ is the number of live births per year and $\delta$ is the death rate per year for individuals who are not infectious. In model (3.2), $\alpha(t)$ and $\delta(t)$ are determined directly from historical datasets, as detailed in §4.2. The constant values $\alpha$ and $\delta$ in model (3.1) are then determined from the work of §5. The value of $\beta$ is always held constant, as equilibrium information about model (3.1) contributes significantly to the §5 approach to estimating autonomous model parameter values. These equilibrium computations appear soon in §3 and require constant values of $\alpha$ and $\delta$.

The estimated *per capita* death rate for infectious individuals is given by $\epsilon$, the average rate at which infected individuals recover is $\gamma$, and $\mu = \gamma + \epsilon$ is the overall rate of loss from $I(t)$.

As a note, some models write $\epsilon = \delta +$ (deaths due to smallpox). We agree that $\epsilon > \delta$ yet compute values of $\delta$ and $\delta(t)$ from historical datasets and compute $\epsilon$ from government-provided smallpox information, as will be shown in §§4.1 and 4.2. Due to these distinct parameter estimation approaches, $\epsilon$ is best described separately from $\delta$ or $\delta(t)$ in models (3.1) and (3.2).

In this model, all newborns are susceptible to smallpox, and disease is transmitted via mass action incidence. The smallpox incubation period, typically 10–14 days [13], is neglected, regarding these individuals as remaining in the $S$ compartment: people move to $I$ only when they become infectious. Vaccination is not considered in this article because the large oscillations in our datasets occurred before extensive vaccine campaigns. This is one of the simplest possible compartmental models with demographic effects, yet its pairing with historical data shows fascinating insight into oscillation and determination of parameter values.

When parameters are held constant, the system has two equilibrium points: a disease-free equilibrium $((\alpha/\delta), 0, 0)$ and an endemic equilibrium

$$
(S^{\star}, I^{\star}, R^{\star}) = \left( \frac{\mu}{\beta}, \frac{\alpha\beta - \delta\mu}{\beta\mu}, \frac{\gamma(\alpha\beta - \delta\mu)}{\beta\delta\mu} \right).
$$

Stability analysis shows that solutions approach the disease-free equilibrium in the case that $\alpha\beta < \delta\mu$. In the case that $\alpha\beta > \delta\mu$, the endemic equilibrium is asymptotically stable, as shown by the eigenvalues provided below and in equation (3.4). Note that on the left-hand side of the inequalities involving $\alpha\beta$ and $\delta\mu$, $\alpha$ indicates replenishment of the susceptible population and $\beta$ determines disease incidence. On the right-hand sides of the inequalities, all terms show removal of people from the infectious compartment ($\mu$) or from the population as a whole ($\delta$), in both cases decreasing opportunity for infection to spread. Therefore, the interpretation of the model matches the mathematical analysis

regarding which equilibrium is favoured, which depends on relative parameter values. Furthermore, $\mathcal{R}_0$ for this system is computed [14] to be

$$\mathcal{R}_0 = \frac{\alpha\beta}{\delta\mu},$$

(3.3)

and $\mathcal{R}_0 > 1$ corresponds exactly to the case when the endemic equilibrium is stable.

In some cases, approach to the stable endemic equilibrium is via damped oscillations. Note that the relevant eigenvalues of the linearized system are $-\delta$ and

$$\frac{-\alpha\beta \pm \sqrt{\alpha^2\beta^2 - 4\alpha\beta\mu^2 + 4\delta\mu^3}}{2\mu},$$

(3.4)

where $\delta > 0$, $\alpha > 0$, $\beta > 0$ and $\mu > 0$, indicating that all eigenvalues have negative real part. Damped oscillations occur when

$$\alpha^2\beta^2 - 4\alpha\beta\mu^2 + 4\delta\mu^3 < 0,$$

with the period $T$ of damped oscillations, that is, the interepidemic interval, given by

$$T = \frac{2\pi}{\sqrt{-(\alpha^2\beta^2)/(4\mu^2) + \alpha\beta - \delta\mu}}.$$

(3.5)

While $\beta$ is not easily computed from epidemiological data, the quantity $\mathcal{R}_0$ is regularly estimated for outbreaks. Substituting (3.3) into the square root in (3.5), the interepidemic interval is

$$T = \frac{2\pi}{\sqrt{-(1/4)\,\mathcal{R}_0^2\,\delta^2 + (\mathcal{R}_0 - 1)\,\delta\mu}}.$$

Parameter values for smallpox shed light on when to expect oscillatory versus asymptotic approach to the endemic equilibrium. In §4, three historical datasets, along with biological smallpox information, are used to determine realistic parameter value ranges. In §5, multiple analyses of periodicity in the historical demographic data further narrow down the possible combinations of parameter values for the autonomous model. Finally, in §6, we summarize results and discuss insights into historical data based on model and data analysis.

# 4. Data

## 4.1. Smallpox-specific parameters

Parameter ranges for $\mathcal{R}_0, \mu, \gamma$ and $\epsilon$ are computed using studies from multiple smallpox outbreaks. Values of $\mathcal{R}_0$ have been estimated to be 3–5 in developing countries before the global eradication campaign [15], 3.5–6 if a contemporary outbreak were to occur [16], and 5–7 in files from the Centers for Disease Control and Prevention (CDC) [17]. Combining ranges, $\mathcal{R}_0 \approx 3 - 7$.

Computing $\mu$, $\gamma$ and $\epsilon$ starts with the average infectious period. The CDC lists several phases of smallpox infection, along with how contagious an infected person is during each phase [13]. Taking these sequentially, smallpox is most contagious for approximately 14 days, and the total contagious period could be as much as 24 days. This means individuals remain in the $I$ compartment 14 to 24 days; on an annual time scale, individuals remain in $I$ for $14/365$ to $24/365$ years. Therefore, $\mu \in [(365/24), (365/14)]$. About 30% of infected individuals historically have died of smallpox [18], signifying that about 70% of the individuals who leave $I$ subsequently move to the $R$ compartment. Therefore, set $\epsilon = 0.3\mu$ and $\gamma = 0.7\mu$, or set $\epsilon \approx 0.3\mu$ and $\gamma = \mu - \epsilon$. A word of caution to the modeller: permitting CDC values for infection length to be used for $\gamma$ alone, that is letting $\gamma \in [(365/24), (365/14)]$, and simultaneously permitting $\epsilon$ to equal an additional 30% of the value of $\gamma$, results in a combined removal rate from $I$ that is higher than indicated by demographic data.

## 4.2. Demographically determined parameters

Three datasets are analysed in this paper: Hida, Japan (1771–1851), British India (1870–1920) and Sweden (1774–1800). Hida data come from [19]: population, deaths overall, and deaths from smallpox, each

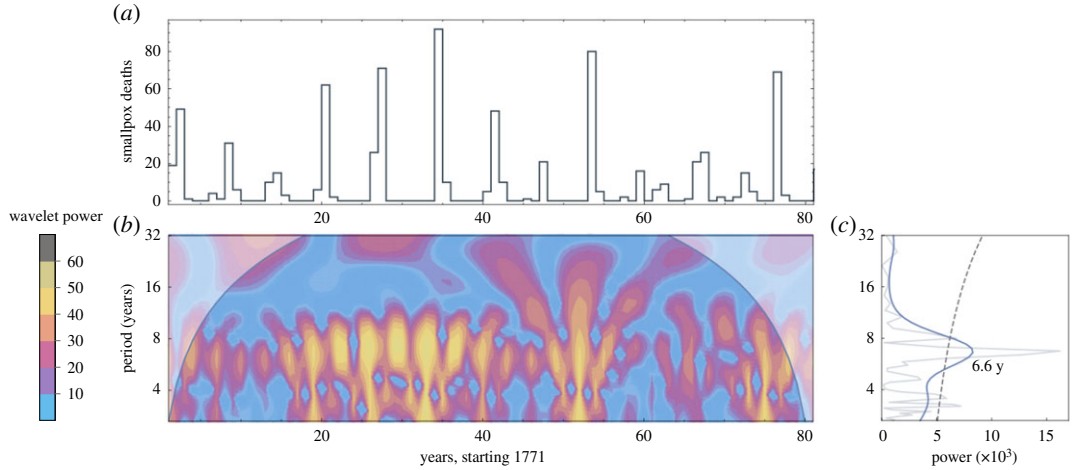

**Figure 1.** Data for Hida, Japan. (*a*) Deaths per year, 1771–1852. (*b*) Derivative of Gaussian wavelet analysis of data in (*a*). The upper corner regions represent power outside the 95% confidence interval. (*c*) Thin grey line shows the Fourier spectrum; thick blue line shows the global wavelet spectrum; and black dashed line shows 95% confidence interval.

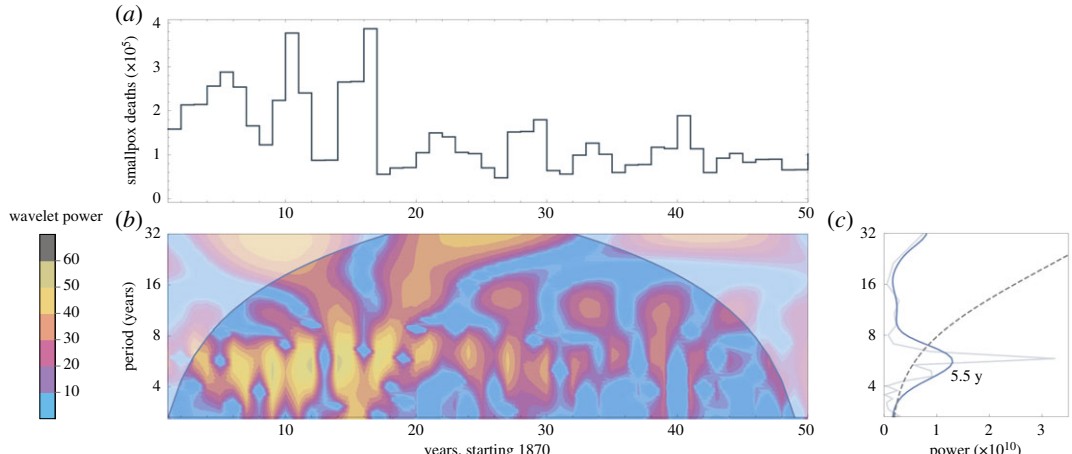

**Figure 2.** Data for British India. (*a*) Deaths per year, 1870–1920. (*b*) Derivative of Gaussian wavelet analysis of data in (*a*). The upper corner regions represent power outside the 95% confidence interval. (*c*) Thin grey line shows the Fourier spectrum; thick blue line shows the global wavelet spectrum; and black dashed line shows 95% confidence interval.

computed annually. Annual numbers of smallpox deaths for British India and Sweden are reported in [20,21], respectively. Total annual deaths and total annual population values for both British India and Sweden come from [22]. The regions' smallpox death data appear visually in figures 1*a*, 2*a* and 3*a*. Each dataset for smallpox deaths is accompanied by its wavelet power spectrum in (*b*) and its global wavelet spectrum and Fourier spectrum in (*c*). The graph in (*c*) includes the dominant period for the data: 6.6 years for Hida, 5.5 years for British India and 5.2 years for Sweden.

To compute $\delta$ for each year $t$, smallpox deaths are subtracted from total deaths for that year, and the resulting number is divided by the population size $P_t$ that year:

$$\delta_t = \frac{(\text{total deaths})_t - (\text{smallpox deaths})_t}{P_t}. \tag{4.1}$$

To compute $\alpha$ for each year, population change $P_{t+1} - P_t$ is set equal to the difference between births and deaths. Rearranging gives

$$\alpha_t = P_{t+1} - P_t + (\text{total deaths})_t. \tag{4.2}$$

Note that $\alpha_t$ and $\delta_t$ are the sole demographically determined parameters in this model. Therefore, population irregularities such as high death counts due to famine are encapsulated in the values of $\alpha_t$ and $\delta_t$. By contrast, the value of $\epsilon$ draws from on-average global smallpox estimates, as shown in §4.1.

**6**

royalsocietypublishing.org/journal/rsos

_R. Soc. open sci._ **7**: 191187

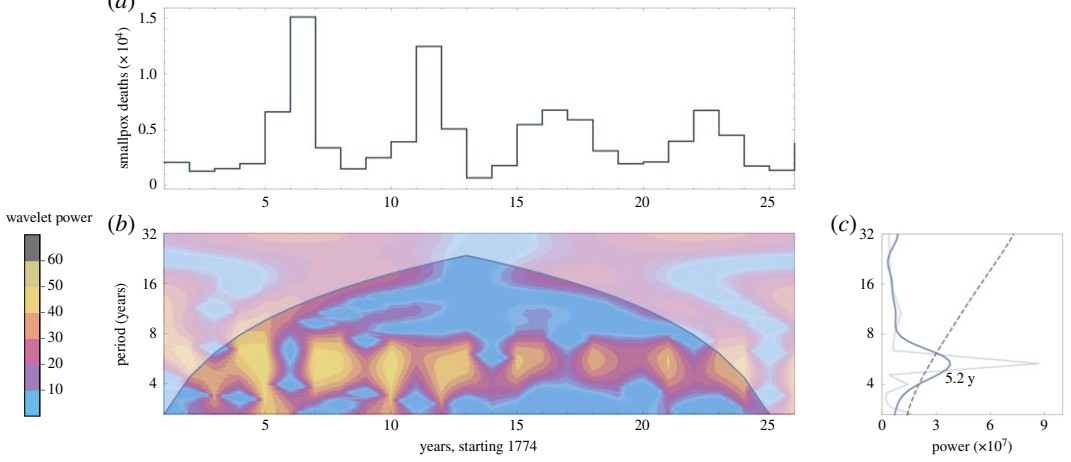

**Figure 3.** Data for Sweden. (_a_) Deaths per year, 1774–1800. (_b_) Derivative of Gaussian wavelet analysis of data in (_a_). The upper corner regions represent power outside the 95% confidence interval. (_c_) Thin grey line shows the Fourier spectrum; thick blue line shows the global wavelet spectrum; and black dashed line shows 95% confidence interval.

This means that any periodicity of smallpox deaths observed in model simulations results from the model and the demographic data; periodicity is not predetermined by a periodically forced parameter such as $\beta$ or $\epsilon$.

To distinguish notation for demographic parameter values: $\alpha_t$ and $\delta_t$ are the discrete year-by-year values computed as shown above. When using model (3.2) for simulations, $\alpha_t$ and $\delta_t$ are linearly interpolated to produce $\alpha(t)$ and $\delta(t)$. This prevents discontinuous inputs to the differential equations of model (3.2), which improves the stability of numerical results.

Full datasets and programs used to convert data to $\alpha_t$ and $\delta_t$ appear in this article's electronic supplementary material.

## 4.3. The infection parameter

The value of $\beta$ does not draw as directly from historical or epidemiological data. Instead, $\mathcal{R}_0$ is instrumental in determining the range of viable values for $\beta$. Equation (3.3) is rewritten as

$$\beta = \frac{\mathcal{R}_0 \, \delta \, \mu}{\alpha}, \tag{4.3}$$

and values for $\mathcal{R}_0$, $\delta$, $\mu$ and $\alpha$ are substituted to compute the full range of possible $\beta$ values. In particular, to compute the smallest possible $\beta$, the minimum values of $\mathcal{R}_0$, $\delta$ and $\mu$ are substituted into equation (4.3), along with the maximum value of $\alpha$. The largest possible $\beta$ is computed using the maximum $\mathcal{R}_0$, $\delta$ and $\mu$, and the minimum value of $\alpha$.

Within any single model simulation, the value of $\beta$ is held constant: only demographically determined $\alpha(t)$ and $\delta(t)$ may vary within a simulation. Section 5 shows how data from figures 1–3 can be used to select constant $\beta$ values for use in models (3.1) and (3.2).

Combined parameter values appear in table 1.

# 5. Connecting model simulations with data

The phase portraits in figure 4 encapsulate the differences between the autonomous and demographically forced models. The model (3.1) mathematical analysis in §3 details the damped oscillation and periodicity expected for any autonomous set of parameters, resulting in the consistent and smooth phase space behaviour in the left image of figure 4. The demographic forcing of model (3.2) permits sustained and irregular oscillation, as in the right image of figure 4; such trajectories provide the opportunity to calibrate parameter values using a breadth of historical data.

The following three sections detail our approach to computing model parameters using historical data. In §5.1, a wide range of $\mathcal{R}_0$ and $\mu$ values are tested against demographic data to determine

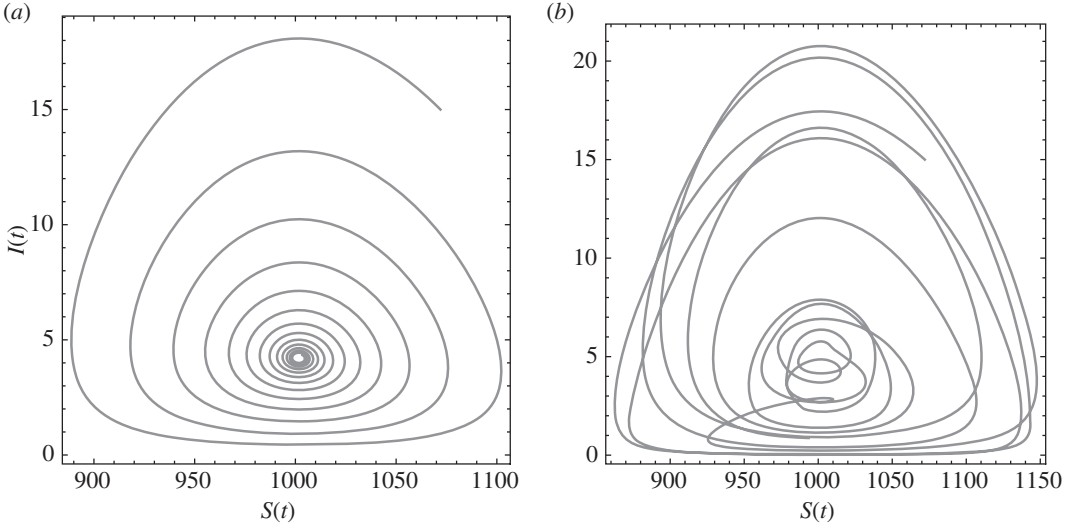

**Figure 4.** Sample phase portraits for Hida, Japan (*a*) damped oscillation using the autonomous model (3.1) with $\alpha = \alpha_{\text{med}} = 96.5$, $\delta = \delta_{\text{med}} = 0.028$; (*b*) sustained oscillation using the demographically forced model (3.2). The parameters $\mu = 16$, $R_0 = 3.05$ for both. The infection parameter $\beta$ is computed using equation (4.3).

**Table 1.** Summary of parameter values for models (3.1) and (3.2).

| parameter | interpretation | values modelled | units |
|---|---|---|---|
| $\mathcal{R}_0$ | basic reproduction number | [3, 7] | – |
| $\mu$ | combined removal rate from $I$ compartment | $\left[\frac{365}{24}, \frac{365}{14}\right]$ | $\text{yr}^{-1}$ |
| $\gamma$ | recovery rate for infected individuals | $\left[0.7 \times \frac{365}{24}, 0.7 \times \frac{365}{14}\right]$ | $\text{yr}^{-1}$ |
| $\epsilon$ | death rate for infected individuals | $\left[0.3 \times \frac{365}{24}, 0.3 \times \frac{365}{14}\right]$ | $\text{yr}^{-1}$ |
| $\delta$ | death rate for non-infected individuals (computed annually as $\delta_t$) | Hida: [0.014, 0.12] <br> British India: [0.036, 0.042] <br> Sweden: [0.021, 0.032] | $\text{yr}^{-1}$ |
| $\alpha$ | births (computed annually as $\alpha_t$) | Hida: [46, 136] <br> British India: $[8.8 \times 10^6, 1.2 \times 10^7]$ <br> Sweden: $[2.6 \times 10^4, 6.9 \times 10^4]$ | $\text{people} \cdot \text{yr}^{-1}$ |
| $\beta$ | infectivity parameter | Hida: [0.005, 0.47] <br> British India: $[1.6 \times 10^{-7}, 8.7 \times 10^{-7}]$ <br> Sweden: $[1.7 \times 10^{-5}, 2.2 \times 10^{-4}]$ | $\text{people}^{-1} \cdot \text{yr}^{-1}$ |

($\mathcal{R}_0^*$, $\mu^*$) pairs that produce periodicity closest to the dominant period in the data, as computed in figures 1–3. A single value of $\beta^*$ is then computed directly from $\mathcal{R}_0^*$ and $\mu^*$.

Section 5.2 supports the work in §5.1 by computing a set of likely values for $\beta^*$. This is done by fixing a pair ($\mathcal{R}_0^*$, $\mu^*$) based on §5.1, running the demographically forced model (3.2) for each of a wide range of possible $\beta$ values, computing the power spectrum of the output for each model run, and determining the corresponding 95% confidence interval. These results are compared with both the data periodicity determined in §4.2 and the $\beta^*$ value from §5.1.

Section 5.3 starts with paired values of $\beta^*$ and $\mu^*$ computed in §5.1 and supported by §5.2. For a wide-ranging set of values of $\alpha$ and $\delta$, oscillation periods for the autonomous model (3.1) are computed. These periods are compared with the spectral analysis of §4.2.

The steps are detailed below.

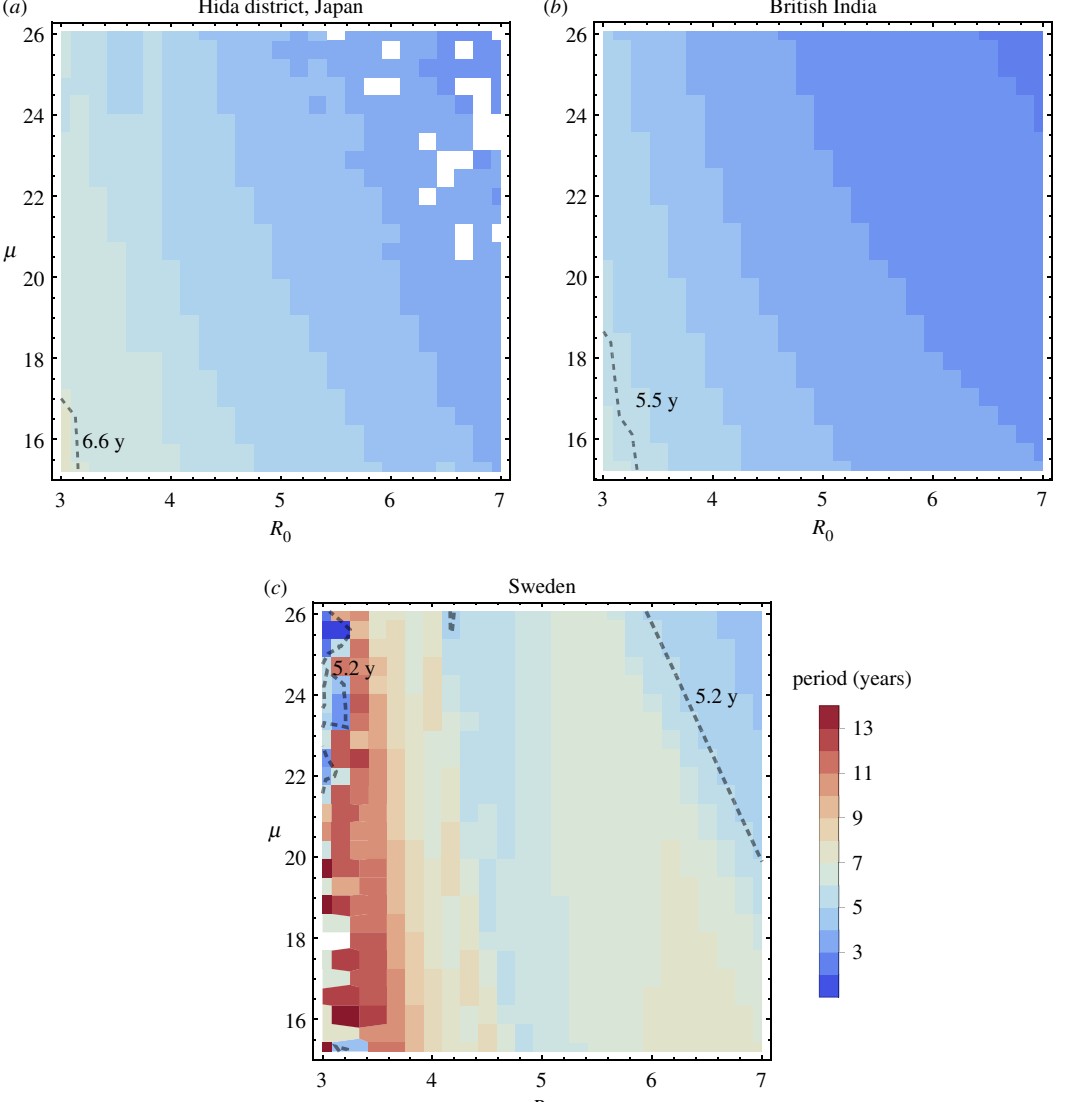

**Figure 5.** Simulated outbreak periods of the demographically forced model over a range of values of the parameters $\mathcal{R}_0$ and $\mu$. The colour of each pixel in the $25 \times 25$ grid represents the oscillation period of the demographically forced model with time-dependent $\alpha(t)$ and $\delta(t)$ and all other parameters held constant. For each pixel, $\mathcal{R}_0$ and $\mu$ are shown as values on the axes, and a corresponding constant value of $\beta$ is computed using equation (4.3) and median values $\alpha_{\mathrm{med}}$ and $\delta_{\mathrm{med}}$ of the respective demographic data. The black dashed line marks the dominant outbreak period obtained from spectral analysis of outbreak data (figures 1–3). White pixels represent numerical instabilities. (a) $\alpha_{\mathrm{med}} = 96.5$, $\delta_{\mathrm{med}} = 0.028$, (b) $\alpha_{\mathrm{med}} = 1.1 \times 10^7$, $\delta_{\mathrm{med}} = 0.041$, (c) $\alpha_{\mathrm{med}} = 4.2 \times 10^4$, $\delta_{\mathrm{med}} = 0.025$.

## 5.1. Constraining $(\mathcal{R}_0, \mu)$ pairs using the demographically forced model

This first set of computations locates pairs $(\mathcal{R}_0, \mu)$ that cause the demographically forced model (3.2) to produce the mean period closest to the dominant outbreak period. Several full simulations of the demographically forced model are run, incorporating demographic data $\alpha(t)$ and $\delta(t)$ along with constant values $\mathcal{R}_0$ and $\mu$. For each simulation, the mean period length is computed and compared against the dominant outbreak period determined in §4.2. Results appear in figure 5.

Full biologically viable ranges of $\mathcal{R}_0$ and $\mu$ are tested; these values appear in table 1. For each historical dataset, the ranges of $\mathcal{R}_0$ and $\mu$ form a two-dimensional region, as shown on the axes in figure 5, and this region is divided into a $25 \times 25$ grid of fixed pairs $(\mathcal{R}_0, \mu)$. For each pair $(\mathcal{R}_0, \mu)$ in the grid, a model simulation is run in which $\alpha(t)$ and $\delta(t)$ take on their historical values as determined in §4.2. Their interpolation for smoother integration is completed using the default Interpolation command in Mathematica version 11.

Each model simulation produces a mean period for $I(t)$ oscillations; these mean periods are depicted using colour in the plots of figure 5. Computation of the mean periods shown in figure 5 uses the argrelextrema package from Python's SciPy library to determine local maxima in model output. Within each simulation, times are recorded from each local maximum to the next, and the mean of these times is plotted as the overall period length for the simulation.

Overlaid on each graph in figure 5 is a black dashed line showing the dominant outbreak period for that historical dataset, as computed in the spectral analysis of data shown in figures 1–3. Pairs $(\mathcal{R}_0, \mu)$ along this dashed line are good candidates for inclusion in an autonomous model. For any such selected pair $(\mathcal{R}_0^*, \mu^*)$ the corresponding constant value of $\beta^*$ is computed using equation (4.3) and median values of $\alpha$ and $\delta$.

To summarize, demographic data are used to determine likely parameter values with priority on matching the mean periodicity of the demographically forced model. This approach produces pairs $(\mathcal{R}_0^*, \mu^*)$ and their corresponding $\beta^*$. Section 5.2 considers a wider range of corresponding $\beta$ values for pairs $(\mathcal{R}_0^*, \mu^*)$.

## 5.2. Determining optimal $\beta$ values using the demographically forced model

We next complement and build on §5.1, using a new approach to determine a range of appropriate constant $\beta$ values for each model. These experiments revisit pairs $(\mathcal{R}_0^*, \mu^*)$ computed in §5.1. For each pair and its associated historical dataset, simulations of the demographically forced model are run for each of many possible $\beta$ values. Power spectra of model results are then used to indicate values of $\beta$ that both produce periodicity closest to that observed in historical outbreak data and fall within the 95% confidence interval for the spectral analysis.

Three such results appear in figure 6. For each, a single pair $(\mathcal{R}_0^*, \mu^*)$ is selected from its respective graph in figure 5. The selected pairs $(\mathcal{R}_0^*, \mu^*)$ each lie near the middle of the black dashed line showing dominant outbreak period. Then, for each pair $(\mathcal{R}_0^*, \mu^*)$, a range of values $[\beta_{low}, \beta_{high}]$ is computed using equation (4.3). For $\beta_{low}$, $\mu = \mu^*$, $\mathcal{R}_0 = \mathcal{R}_0^*$, $\alpha$ equals the third quartile of all $\alpha_t$ values for the appropriate dataset (i.e. Hida, British India or Sweden), and $\delta$ equals the first quartile of all $\delta_t$ values. Similarly, $\beta_{high}$ is computed with $\mu = \mu^*$, $\mathcal{R}_0 = \mathcal{R}_0^*$, $\alpha$ equal to the first quartile of $\alpha_t$ values, and $\delta$ equal to the third quartile of $\delta_t$ values.

For our selected pair $(\mathcal{R}_0^*, \mu^*)$ and each of 50 fixed and equally spaced values $\beta \in [\beta_{low}, \beta_{high}]$, the demographically forced model is run using $\alpha(t)$ and $\delta(t)$. The power spectrum is then computed for the output of each of these 50 model simulations.

Figure 6 shows the collection of power spectra for the three geographical regions of study. Each region's graph consists of 50 vertical strips, each a power spectrum, one for each $\beta \in [\beta_{low}, \beta_{high}]$. A black curve encloses outputs in the 95% confidence interval; these are coloured more boldly than outputs outside the 95% confidence interval.

Overlaid on the graphs in figure 6, and computed directly from the datasets shown in figures 1–3, are solid horizontal lines showing the 95% confidence interval band. The dashed horizontal line shows the peak of the global wavelet spectrum. The vertical line overlaid on each figure 6 graph is the value of $\beta^*$ that corresponds to the pair $(\mathcal{R}_0^*, \mu^*)$ used to create the graph, with $\beta^*$ determined via the method in §5.1. With best values of $\beta$ thus tied to values of $\mathcal{R}_0^*$ and $\mu^*$, §5.3 turns toward determining fixed values of $\alpha$ and $\delta$ for use in the autonomous model (3.1).

## 5.3. Determining fixed $\alpha$ and $\delta$ values for the autonomous model

In §5.1, we obtained pairs of parameter values $(\mathcal{R}_0^*, \mu^*)$ and corresponding $\beta^*$ based on simulations of the demographically forced model. In §5.2, we generated a range of $\beta$ values corresponding to each $(\mathcal{R}_0^*, \mu^*)$ pair, again using the demographically forced model. In this section, we investigate how well the autonomous model reproduces outbreak periods with tuned parameters $\mu$ and $\beta$.

This approach centres on equation (3.5), which computes the oscillation period of the autonomous model as a function of $\mu$, $\beta$, and constant-valued $\alpha$ and $\delta$. Results appear in figure 7: titles show fixed values of $\mu^*$ and $\beta^*$, while $\alpha$ and $\delta$ values are labelled on the axes. A colour-based contour map shows the model periods computed in equation (3.5) for each pair $(\alpha, \delta)$ producing oscillation. Overlaid on the contour map are the historical demographic pairs $(\alpha_t, \delta_t)$.

Overlaid on the graphs are a boldface dashed line showing the peak of the global wavelet spectrum, along with dashed lines showing the 95% confidence band of periods, from the spectral analysis shown in figures 1, 2 and 3. Most $(\alpha_t, \delta_t)$ pairs fall within the 95% confidence band. Additionally, most of the data

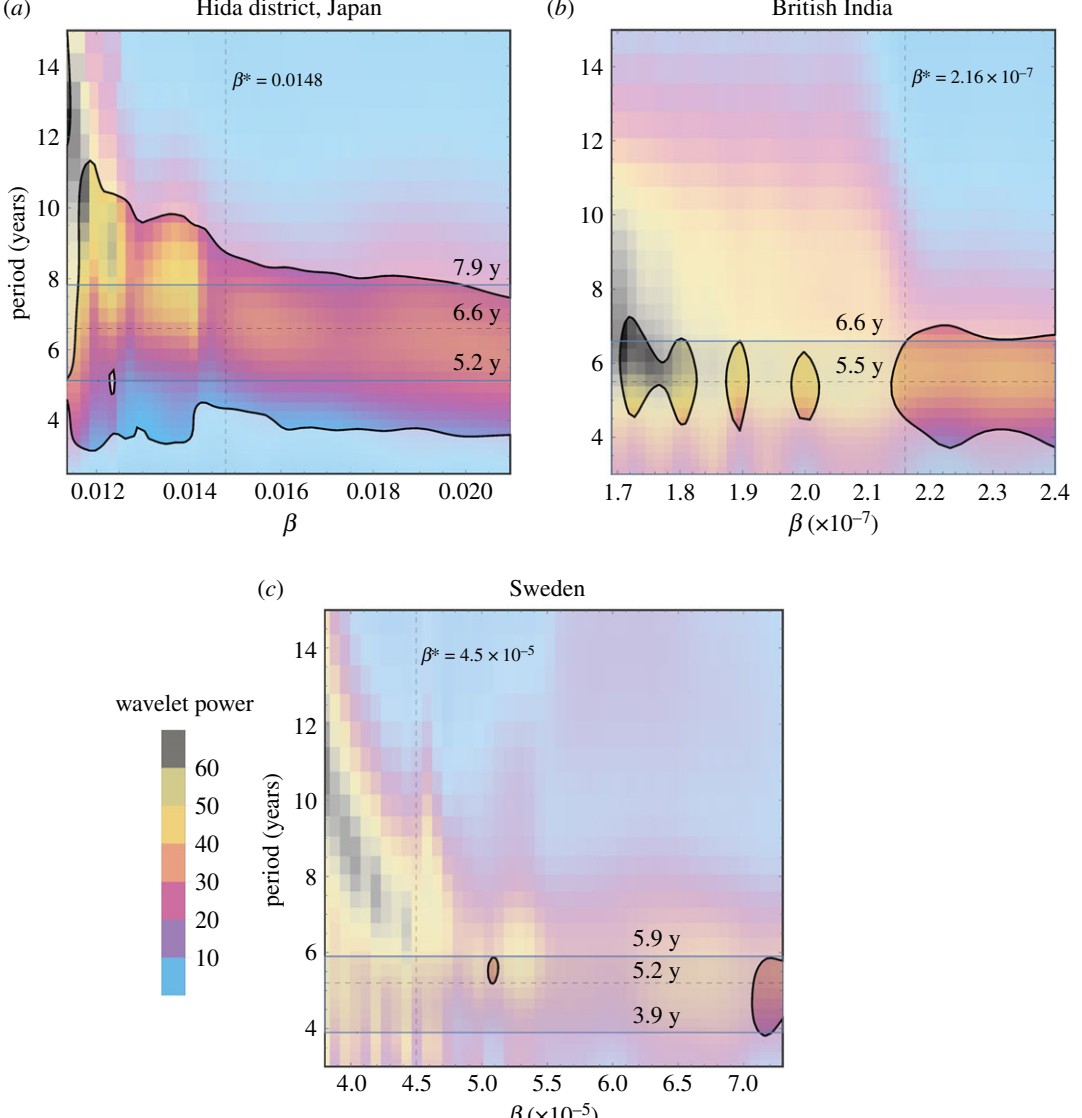

**Figure 6.** Global wavelet power of simulated infection numbers of the demographically forced model using $\alpha(t)$ and $\delta(t)$ from respective countries, for a range of values of $\beta \in [\beta_{low}, \beta_{high}]$. The values of $\mathcal{R}_0$ and $\mu$ are fixed as the pair $(\mathcal{R}_0^*, \mu^*)$ obtained from simulations in figure 5, which are (3.15,16) for Hida, (3.2,16.5) for British India and (3.3,24) for Sweden.

pairs fall within the zone of complex eigenvalues which yield oscillating solutions. Japan's Hida district includes the only data points yielding real eigenvalues and therefore non-oscillating solutions. British India's data pairs all lie within the 95% confidence band.

The analyses in §§5.1, 5.2 and 5.3 show that the simple, autonomous model (3.1) can be calibrated to produce oscillations with periodicity matching that of a historical outbreak. Though such oscillations are damped in an autonomous model, introducing stochasticity via realistic data or even via noise [6] permits the oscillations to extend indefinitely. When that stochasticity arises from real-life data, or from data with values similar to historical data, and when other parameters are selected accordingly, the oscillations remain in a biologically realistic range of periods.

# 6. Discussion

Our results from §§5.1, 5.2 and 5.3 show consistent periodicity between historical data and both the demographically forced and autonomous models. This consistency shows that biological disease information and historical demographic data can produce autonomous parameter values for model (3.1). Resulting simulations of model (3.1) match outbreak periodicity in the short term, with

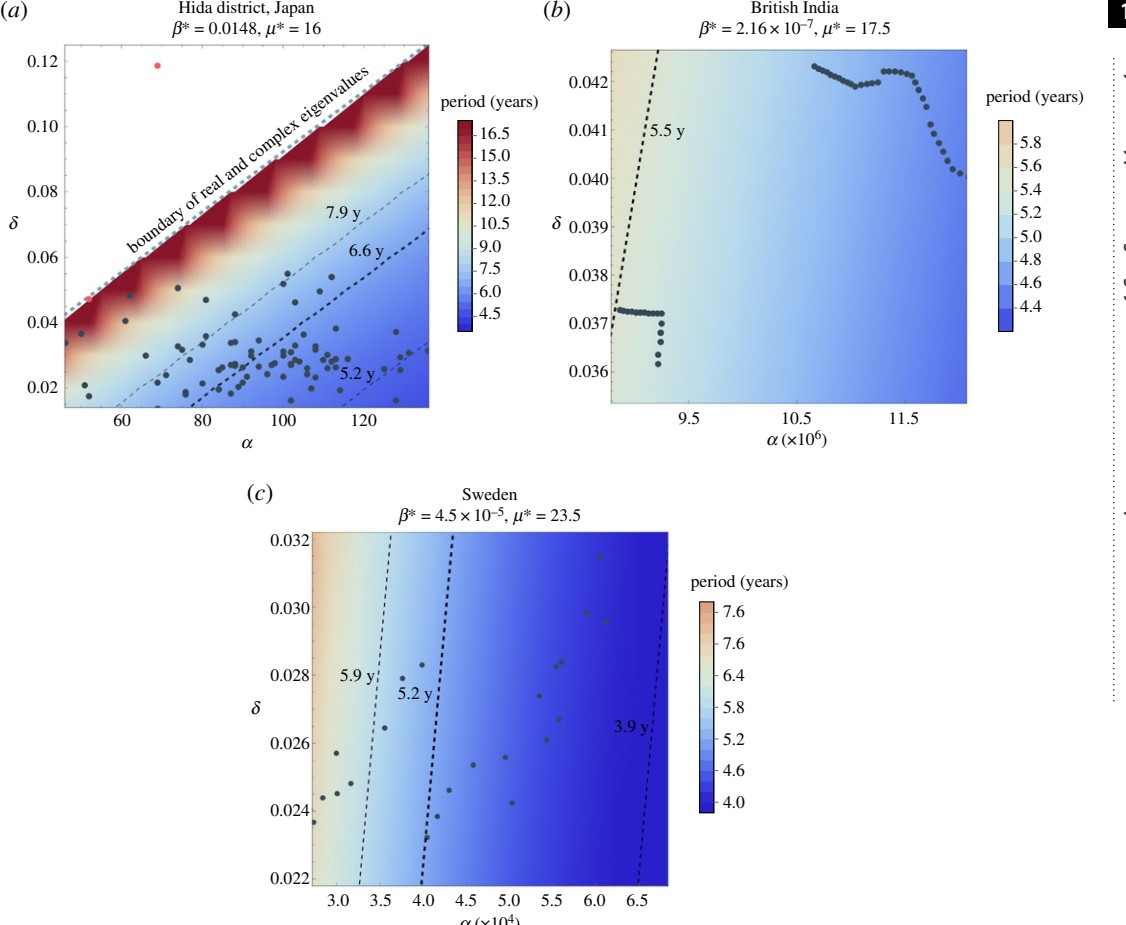

**Figure 7.** Oscillation periods shown in colour as functions of $\alpha$ and $\delta$ as derived in equation (3.5). Actual demographic data of $(\alpha_t, \delta_t)$ are overlaid along with the band of periods above the 95% confidence interval obtained from spectral analysis of outbreak data. The period with maximum global power is marked with a thick dashed line. In the case of Japan's Hida district in (a), the two points marked in red represent famine years, and most of the data points fall in the zone of complex eigenvalues that yield oscillating solutions. The entire demographic data for British India and Sweden fall within the zone of complex eigenvalues.

oscillations damped because the model is autonomous. The corresponding demographically forced model (3.2) can match outbreak periodicity in the long term, including sustained oscillations, without the need of a tuned periodic forcing or delay term to impose the oscillation period.

Autonomous values of $\alpha$ and $\delta$ are determined via an analysis of data. Median values of each, computed from demographic data, are a good first approximation for use in the autonomous model. The results of §5.3 provide more precisely tuned combinations of $\alpha$ and $\delta$ that best match the periodicity of the historical outbreak.

Values of $\mathcal{R}_0$ and $\mu$ are biologically determined from information on smallpox. Web information is readily available from the CDC [13,17], for instance. Because $\mathcal{R}_0$ is unitless and $\mu$ is a rate, their values change minimally across different outbreaks. Medical care differences and public health approaches have some effect on $\mathcal{R}_0$ and $\mu$, but their values are well represented by ranges provided by the CDC and other national or international health centres.

Compared with all other parameters in models (3.1) and (3.2), the infection parameter $\beta$ is the least easy to estimate from datasets or biological disease information. The value of $\beta$ depends on population size and density, as well as various dynamical aspects of disease transmission such as types and frequencies of interactions. Indeed, the units of $\beta$ show dependence on the number of people in the population. Therefore, the computational work in §5 is required to systematically narrow down a range of values of $\beta$ based on model outputs and agreement with outbreak and demographic data.

One outcome of our analysis is an overlapping set of values relating directly to the infection parameter. While the numbers found for $\beta$ itself have very different orders of magnitude, rescaling by population size $N = S + I + R$ produces values $\beta_0 = N\beta$ that can be compared; here, $N$ is the median

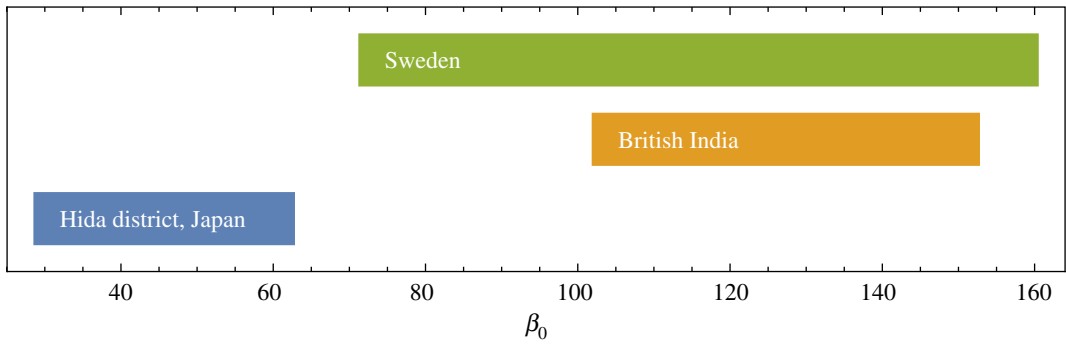

**Figure 8.** Range of values of re-scaled $\beta_0 = \beta(S + I + R)$ for Japan, British India and Sweden.

population across all years of data. Figure 8 shows the range of values of $\beta_0 \in [\beta_{\text{low}}, \beta_{\text{high}}]$ for all three countries. It is worthwhile to note the correspondence between the geographical scale of data and the range of values for $\beta_0$.

A second outcome of our analysis is identification of parameter combinations $(\alpha, \delta, \mu, \beta)$ for which the autonomous model best matches the periodicity of historical data. These parameter combinations appear as bold dashed $(\alpha, \delta)$ lines in figure 7 for which $\mu$ and $\beta$ are fixed, with most historical pairs $(\alpha_t, \delta_t)$ lying close to these lines of maximum global power, and almost all historical pairs lying within the 95% confidence interval. This comparison of analysis with data shows that an autonomous model can match periodicity well for the short term. Longer term, historical birth and death rates introduce stochasticity, which sustains oscillations, yet almost always producing oscillation periods that match well with the signal from historical data.

Of further interest are the exceptions to historical $(\alpha_t, \delta_t)$ pairs within the spectral analysis 95% confidence interval. British India has no exceptions, and Sweden's exceptions are few. Hida's exceptions include two years of famine, marked in red on figure 7. Hida is also the only case in which historical $(\alpha_t, \delta_t)$ combinations produce a non-negative square root in equation (3.4). A non-negative square root is a rare occurrence, to the extent that textbook analyses (such as [23]) typically neglect the possibility so that formula analysis is more tractable: for example, the oscillation period then has a more tidy solution.

A third outcome of our analysis is that its success is enhanced by longer datasets covering more homogeneous populations and geographical regions. In particular, the dataset for Hida is 81 years, the dataset for British India is 51 years, and the dataset for Sweden spans only 26 years. Further advantageous for Hida is that it is a relatively small and homogeneous population in a small geographical region. For these reasons, the Hida dataset appears to best lend itself toward this analysis. Hida's peaks of smallpox incidence are clearest, rising the most in comparison with non-peak years and spanning just one to two calendar years. For contrast, while oscillation is noticeable in the British India and Sweden datasets, it is yet true that smallpox incidence in those regions encompasses a much larger geographical area, so that outbreaks spread spatially and therefore take up multiple years apiece. This smooths and spreads out some of the effects of oscillation. Length of dataset affects the oscillation periods modelled using demographic data, such as in figure 5, where the pattern is quite similar in period lengths for Hida and British India, but the smaller Sweden dataset leads to more variation in period lengths as $\mathcal{R}_0$ and $\mu$ change.

These second and third outcomes, combined, support the interest in outlier points in the Hida dataset. Hida's data are well suited to these analyses due to both number of data points and homogeneity of population. As further support for this model, research [19] shows that, aside from famine in two separate years, smallpox was the main contributor to untimely deaths during the years of our dataset. Therefore, the $(\alpha_t, \delta_t)$ pairs lying outside the 95% confidence interval, or even outside the region of parameters for which model (3.1) oscillates for otherwise-appropriate $(\mu, \beta)$ combinations, may hold important information about the presence of smallpox in Hida. By using simple models to emphasize the connections between data and mathematical properties such as oscillation, we can thus highlight historical data points of interest for further study. This cycle, from data to model calibration to model simulation to re-emphasis on data, has the potential to reflect back to us historical data points of special interest.

The fourth and final outcome of our analysis is that periodic forcing, delays and other exogenous mechanisms are not necessary ingredients for generating periodic behaviour in a model. Instead, the

application of demographic forcing alone, via rates of live birth and of death, can produce sustained oscillations with similar periods as observed infection rates.

This further suggests that the underlying dynamical mechanisms in disease outbreaks can be captured well by simple SIR models. Meanwhile, datasets from historical outbreaks worldwide have become widely available for study. We, therefore, call for a renewed interest in relatively simple mathematical models, in conjunction with data from real outbreaks. This combination of models with data will bring new insights into models, especially models involving realistic demographics. In turn, these models can shed light on special or unusual data points, for improved understanding of the relationship between stochastic real-world dynamics and the varying incidence of endemic disease.

Data accessibility. The code and the datasets supporting this article have been uploaded as electronic supplementary material.

Authors' contributions. M.G. estimated parameter values, co-developed and mathematically analysed the model and drafted the manuscript; R.S. estimated parameter values, designed and performed computational analyses and helped draft the manuscript; A.G. collected epidemiological and demographic data, curated information from many sources, carried out a literature review of prior work on periodic disease outbreaks, and provided critical direction and content for the manuscript; C.Y. characterized theoretical conditions in which model oscillations occur, estimated parameter values and initial conditions, and contributed critical intellectual content to the manuscript; K.Z.-V. developed initial model formulation and analytical solutions, collected epidemiological and demographic data, carried out a literature review of prior work on periodic disease outbreaks, and provided critical direction and content for the manuscript. All authors gave final approval for publication and agree to be held accountable for the work performed therein.

Competing interests. We declare we have no competing interest.

Funding. No funding has been received for this article.

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
