## [Reviewer comments · Royal Society Open Science]

Review History

RSOS-191187.R0 (Original submission)

Review form: Reviewer 1 (Fred Brauer)

Is the manuscript scientifically sound in its present form?

Yes

Are the interpretations and conclusions justified by the results?

Yes

Is the language acceptable?

Yes

Do you have any ethical concerns with this paper?

No

Have you any concerns about statistical analyses in this paper?

No

Recommendation?

Accept as is

Comments to the Author(s)

1. In the model (1), I would suggest writing the departure term from I as a sum of terms for natural deaths, disease deaths, and recovery.
2. In the paper I did not see a reference to [5], but I believe it would be appropriate to mention it along with [6].
3. I did not see a reference to [3] and would suggest deleting this reference. On the other hand, there are two references to chapters in the Mathematical Epidemiology, [2] and [7], but the bibliography does not include Mathematical Epidemiology.
4. The paper refers to the paper by Soper in 1929, and I suggest adding that this paper raises an important question but is unsatisfactory because it uses a model that is not properly posed mathematically.

Review form: Reviewer 2**Is the manuscript scientifically sound in its present form?**

Yes

Are the interpretations and conclusions justified by the results?

Yes

Is the language acceptable?

Yes

Do you have any ethical concerns with this paper?

No

Have you any concerns about statistical analyses in this paper?

No

Recommendation?

Major revision is needed (please make suggestions in comments)

Comments to the Author(s)

My major questions are about the starting point: the proposed SIR model. Such a model is very simple. At least, similar models should be cited (e.g. Ecological Modelling 197 (2006) 258–262; Ecological Complexity 17 (2014) 40–45). Also, other causes for oscillatory behavior should be included in the Introduction (e.g. Ecological Complexity 17 (2014) 40–45; Ecological Complexity 31 (2017) 57–63).

The authors should justify the assumption that births occur at a constant rate. Usually, births depend on the population size.

In Eq. (5), T is the period of "damped" oscillations. This should be clearly mentioned. Also, it should be proved that the autonomous model can not support self-sustained oscillations (stable limit-cycles). Hence, the forcing terms are necessary for creating oscillation.

It is not clear the meaning of the colors in the colored plots in Figures 1, 2, 3, and 6. A color bar is missing.

The authors should state that the proposed model is not adequate for describing current smallpox epidemics because a vaccination term is missing.

Can the authors provide mathematical expressions from data for the parameters $\alpha(t)$ and $\delta(t)$ in the forced model? Also, why is β not considered time-dependent too?

The newest reference is from 2008. This should be fixed.

Decision letter (RSOS-191187.R0)

02-Sep-2019

Dear Dr Greer,

The editors assigned to your paper ("Emergence of oscillations in a simple epidemic model with demographic data") have now received comments from reviewers. We would like you to revise your paper in accordance with the referee and Associate Editor suggestions which can be found below (not including confidential reports to the Editor). Please note this decision does not guarantee eventual acceptance.

Please submit a copy of your revised paper before 25-Sep-2019. Please note that the revision deadline will expire at 00.00am on this date. If we do not hear from you within this time then it will be assumed that the paper has been withdrawn. In exceptional circumstances, extensions may be possible if agreed with the Editorial Office in advance. We do not allow multiple rounds of revision so we urge you to make every effort to fully address all of the comments at this stage. If deemed necessary by the Editors, your manuscript will be sent back to one or more of the original reviewers for assessment. If the original reviewers are not available, we may invite new reviewers.

- Data accessibility

It is a condition of publication that all supporting data are made available either as supplementary information or preferably in a suitable permanent repository. The data accessibility section should state where the article's supporting data can be accessed. This section

should also include details, where possible of where to access other relevant research materials such as statistical tools, protocols, software etc can be accessed. If the data have been deposited in an external repository this section should list the database, accession number and link to the DOI for all data from the article that have been made publicly available. Data sets that have been deposited in an external repository and have a DOI should also be appropriately cited in the manuscript and included in the reference list.

If you wish to submit your supporting data or code to Dryad (<http://datadryad.org/>), or modify your current submission to dryad, please use the following link:
<http://datadryad.org/submit?journalID=RSOS&manu=RSOS-191187>

- **Competing interests**

- **Authors' contributions**

- **Acknowledgements**

- **Funding statement**

on behalf of Dr Anna Marciniak-Czochra (Associate Editor) and Mark Chaplain (Subject Editor)
openscience@royalsociety.org

Comments to Author:

Reviewers' Comments to Author:

Reviewer: 1

Comments to the Author(s)

1. In the model (1), I would suggest writing the departure term from I as a sum of terms for natural deaths, disease deaths, and recovery.
2. In the paper I did not see a reference to [5], but I believe it would be appropriate to mention it along with [6].
3. I did not see a reference to [3] and would suggest deleting this reference. On the other hand, there are two references to chapters in the Mathematical Epidemiology, [2] and [7], but the bibliography does not include Mathematical Epidemiology.
4. The paper refers to the paper by Soper in 1929, and I suggest adding that this paper raises an important question but is unsatisfactory because it uses a model that is not properly posed mathematically.

Reviewer: 2

Comments to the Author(s)

My major questions are about the starting point: the proposed SIR model. Such a model is very simple. At least, similar models should be cited (e.g. Ecological Modelling 197 (2006) 258–262; Ecological Complexity 17 (2014) 40–45). Also, other causes for oscillatory behavior should be included in the Introduction (e.g. Ecological Complexity 17 (2014) 40–45; Ecological Complexity 31 (2017) 57–63).

The authors should justify the assumption that births occur at a constant rate. Usually, births depend on the population size.

In Eq. (5), T is the period of "damped" oscillations. This should be clearly mentioned. Also, it should be proved that the autonomous model can not support self-sustained oscillations (stable limit-cycles). Hence, the forcing terms are necessary for creating oscillation.

It is not clear the meaning of the colors in the colored plots in Figures 1, 2, 3, and 6. A color bar is missing.

The authors should state that the proposed model is not adequate for describing current smallpox epidemics because a vaccination term is missing.

Can the authors provide mathematical expressions from data for the parameters $\alpha(t)$ and $\delta(t)$ in the forced model? Also, why is β not considered time-dependent too?

The newest reference is from 2008. This should be fixed.

Author's Response to Decision Letter for (RSOS-191187.R0)

See Appendix A.

RSOS-191187.R1 (Revision)

Review form: Reviewer 2

Is the manuscript scientifically sound in its present form?

Yes

Are the interpretations and conclusions justified by the results?

Yes

Is the language acceptable?

Yes

Do you have any ethical concerns with this paper?

No

Have you any concerns about statistical analyses in this paper?

No

Recommendation?

Accept with minor revision (please list in comments)

Comments to the Author(s)

The references should be revised. There are some typos.

Decision letter (RSOS-191187.R1)

02-Jan-2020

Dear Dr Greer,

It is a pleasure to accept your manuscript entitled "Emergence of oscillations in a simple epidemic model with demographic data" in its current form for publication in Royal Society Open Science. The comments of the reviewer(s) who reviewed your manuscript are included at the foot of this letter.

on behalf of Dr Anna Marciniak-Czochra (Associate Editor) and Mark Chaplain (Subject Editor)
openscience@royalsociety.org

Reviewer comments to Author:
Reviewer: 2

Comments to the Author(s)
The references should be revised. There are some typos.

Appendix A

To the editors and reviewers:

Thank you for your prompt and thoughtful reviews of our paper “Emergence of oscillations in a simple epidemic model with demographic data.” We have addressed each reviewer comment, as described below, and enthusiastically resubmit this work.

Reviewer: 1

Comments to the Author(s)

1. In the model (1), I would suggest writing the departure term from I as a sum of terms for natural deaths, disease deaths, and recovery.

We appreciate this suggestion and spent a significant amount of time considering how or if to implement it as stated. Indeed, earlier drafts of our models and paper used this exact approach, writing “delta + (smallpox-only death term)” instead of the current “epsilon” that combines both. However, as we wrote, and again as we revised in response to this comment by Reviewer 1, we found that the very different approaches to data collection and parameter estimation for delta(t) and epsilon meant that we should describe these parameters entirely separately. Therefore, instead of changing parameter names, we took seriously the possibility that readers would wonder about our reasoning, and so we inserted into the paper an explanation about our approach to parameter naming. This appears in Section 3, shortly after the equations for Model (2).

2. In the paper I did not see a reference to [5], but I believe it would be appropriate to mention it along with [6].

We thank this reviewer for checking throughout for our references. Reference [5] appears halfway through the second paragraph in Section 4.1; this reference tells us that 3 out of 10 people who had smallpox died. Reference [6] appears in the first paragraph of Section 4.1 and provides values of R_0 for several past outbreaks of smallpox. We therefore keep both [5] and [6] in the reference section and do not include [5] where [6] appears. In the new version of the article, the reference that had been numbered [6] is now [4].

3. I did not see a reference to [3] and would suggest deleting this reference. On the other hand, there are two references to chapters in the Mathematical Epidemiology, [2] and [7], but the bibliography does not include Mathematical Epidemiology.

Again, our thanks to this reviewer for reference suggestions. We have implemented the reviewer’s suggestion of including a dedicated reference to the book Mathematical Epidemiology, using this in place of the former [2] and [7], which instead referenced individual chapters. Because reference [3] appears in paragraph 7 of the Discussion (“Of further interest...”), we have not deleted [3].

4. The paper refers to the paper by Soper in 1929, and I suggest adding that this paper raises an important question but is unsatisfactory because it uses a model that is not properly posed mathematically.

We appreciate this comment and have responded by de-emphasizing Soper's historical role in Section 2, then by stating explicitly that other approaches, notably compartmental modeling, are (rightfully) preferred and have been for nearly a century.

Reviewer: 2

Comments to the Author(s)

My major questions are about the starting point: the proposed SIR model. Such a model is very simple. At least, similar models should be cited (e.g. Ecological Modelling 197 (2006) 258–262; Ecological Complexity 17 (2014) 40–45). Also, other causes for oscillatory behavior should be included in the Introduction (e.g. Ecological Complexity 17 (2014) 40–45; Ecological Complexity 31 (2017) 57–63).

We appreciate this reviewer's suggestions about references. We have included these as well as additional references in Section 2, using the references to lead up to introducing our model.

The authors should justify the assumption that births occur at a constant rate. Usually, births depend on the population size.

We thank you for the suggestion that we comment further on the constant birth rate assumption of Model (1). We have updated the description of parameters to emphasize that we indeed use demographic data, as suggested by Reviewer 2, to determine $\alpha(t)$ and $\delta(t)$ in Model (2), and then use the many methods of Section 5 to move toward reasonable constant values of α and δ for Model (1), for the case in which we wish to use as simple a model as possible.

In Eq. (5), T is the period of "damped" oscillations. This should be clearly mentioned. Also, it should be proved that the autonomous model can not support self-sustained oscillations (stable limit-cycles). Hence, the forcing terms are necessary for creating oscillation.

We are happy to make these changes. We have added that oscillations are damped, both just before Equation (4), and just before Equation (5). Additionally, we expanded our explanation of the endemic equilibrium's stability, adding text shortly after stating the endemic equilibrium, then showing that the real parts of all eigenvalues are negative for the corresponding linear, autonomous, system.

It is not clear the meaning of the colors in the colored plots in Figures 1, 2, 3, and 6. A color bar is missing.

Thank you: we agree that a color bar for wavelet power in Figures 1, 2, 3, and 6 is an improvement, and we have now included this. We believe this is the only change to these figures intended by this reviewer, but for clarity, we also draw attention to the caption descriptions of the gray, blue, and dashed lines in Figure (c).

The authors should state that the proposed model is not adequate for describing current smallpox epidemics because a vaccination term is missing.

We thank Reviewer 2 for suggesting we comment on our model not including vaccination. We made this change in Section 3 as part of describing the parameters and assumptions in our model.

Can the authors provide mathematical expressions from data for the parameters $\alpha(t)$ and $\delta(t)$ in the forced model? Also, why is β not considered time-dependent too?

Thank you for this comment: we realize we can create a clearer connection between raw data and our model. The formulas for determining α_t and δ_t appear in Equations (6) and (7) of our manuscript. The numerical values at continuous time $\alpha(t)$ and $\delta(t)$, as needed in our simulations, are computed using linear interpolation between the demographic data points. The raw data appear in Supplementary Files (see the list of all pertinent files below).

Data files:

Japan_Hida_year_alphatot_delta.dat

smallpox_totaldeaths_HidaJapan.dat

British_India_alphatot_vs_delta.dat

smallpox_totaldeaths_BritishIndia.dat

Sweden_year_alpha_delta.dat

smallpox_totaldeaths_Sweden.dat

The linear interpolation is done using the `interp1d` library in python, with the default setting for linear. See relevant program files and line numbers below.

`sir_module_beta_alphadeltaForced.py` (lines 29, 30)

`sir_module.py` (lines 33, 34, 48, 49)

We have added a comment at the end of section 4.2, directing readers to this supplemental material.

To address the reviewer's comment on why β is not time-dependent: we agree this is important to discuss and therefore provide reasoning for this in Section 3, just below the definitions of Models (1) and (2).

The newest reference is from 2008. This should be fixed.

We agree entirely that there should be more recent references, and we have remedied this.